# Post-spreading Basalts from the Nanyue Seamount: Implications for the Involvement of Crustal- and Plume-Type Components in the Genesis of the South China Sea Mantle

**Hao Zheng** [1,2] , **Li-Feng Zhong** [1,2,*], **Argyrios Kapsiotis** [1,2], **Guan-Qiang Cai** [3,*],
**Zhi-Feng Wan** [1,2] **and Bin Xia** [1,2]

1   School of Marine Sciences, Sun Yat-sen University, Guangzhou 510275, China;
    zhengh66@mail.sysu.edu.cn (H.Z.); kapsiotis@mail.sysu.edu.cn (A.K.); wanzhif@mail.sysu.edu.cn (Z.-F.W.);
    xiabin@mail.sysu.edu.cn (B.X.)
2   Southern Marine Science and Engineering Guangdong Laboratory, Zhuhai 519082, China
3   Guangzhou Marine Geological Survey, Guangzhou 510760, China
*   Correspondence: zhonglf9@mail.sysu.edu.cn (L.-F.Z.); caiguanqiang@sina.com (G.-Q.C.);
    Tel.: +86-0208-411-3721 (L.-F.Z.); +86-0208-202-2732 (G.-Q.C.)

**Abstract:** Fresh samples of basalts were collected by dredging from the Nanyue intraplate seamount in the Southwest sub-basin of the South China Sea (SCS). These are alkali basalts displaying right-sloping, chondrite-normalized rare earth element (REE) profiles. The investigated basalts are characterized by low Os content (60.37–85.13 ppt) and radiogenic $^{187}Os/^{188}Os$ ratios (~0.19 to 0.21). Furthermore, $^{40}Ar/^{39}Ar$ dating of the Nanyue basalts showed they formed during the Tortonian (~8.3 Ma) and, thus, are products of (Late Cenozoic) post-spreading volcanism. The Sr–Nd–Pb–Hf isotopic compositions of the Nanyue basalts indicate that their parental melts were derived from an upper mantle reservoir possessing the so-called Dupal isotopic anomaly. Semiquantitative isotopic modeling demonstrates that the isotopic compositions of the Nanyue basalts can be reproduced by mixing three components: the average Pacific midocean ridge basalt (MORB), the lower continental crust (LCC), and the average Hainan ocean island basalt (OIB). Our preferred hypothesis for the genesis of the Nanyue basalts is that their parental magmas were produced from an originally depleted mantle (DM) source that was much affected by the activity of the Hainan plume. Initially, the Hainan diapir caused a thermal perturbation in the upper mantle under the present-day Southwest sub-basin of the SCS that led to erosion of the overlying LCC. Eventually, the resultant suboceanic lithospheric mantle (SOLM) interacted with OIB-type components derived from the nearby Hainan plume. Collectively, these processes contributed crustal- and plume-type components to the upper mantle underlying the Southwest sub-basin of the SCS. This implies that the Dupal isotopic signature in the upper mantle beneath the SCS was an artifact of in situ geological processes rather than a feature inherited from a Southern Hemispheric, upper mantle source.

**Keywords:** basalt; seamount; Hainan plume; South China Sea; Dupal anomaly

## 1. Introduction

A number of Cenozoic extensional basins occur along the eastern continental margin of Eurasia, extending over $5.5 \times 10^3$ km from northeastern Siberia to Indochina. In eastern China, voluminous products of Cenozoic extensional volcanism are systematically scattered along the coastal regions and adjacent offshore shelf from the Heilongjiang province in the north to the Hainan Island/Leizhou

Peninsula and the South China Sea (SCS) in the south [1]. Among these, the Late Cenozoic postspreading basalts from the intraplate seamounts of the SCS are generally characterized by: (i) ocean island basalt (OIB)-type incompatible element distributions, (ii) depleted to moderate Sr–Nd isotopic compositions, and (iii) high $^{208}$Pb/$^{204}$Pb and $^{207}$Pb/$^{204}$Pb ratios for a given $^{206}$Pb/$^{204}$Pb [2–5]. These isotopic signatures are commonly reported from basalts produced by melting of upper mantle domains with a Dupal-like isotopic anomaly (i.e., positive deviation of $^{207}$Pb/$^{204}$Pb and $^{208}$Pb/$^{204}$Pb ratios from the Northern Hemisphere reference line (NHRL)) [6]. Such upper mantle reservoirs reside below an almost linear belt in the Southern latitudinal Hemisphere of our planet. This upper mantle belt stretches from the South Atlantic Ocean (at ~40°S latitude) eastward all the way into the South Indian Ocean, and it is regarded as a long-term petrological aspect of the asthenosphere beneath most of Gondwana [7–9]. Therefore, the origin of the seamount-forming basalts of the SCS, a marginal basin of the Northern latitudinal Hemisphere, is still enigmatic.

In the last few decades the nature of the upper mantle underlying the SCS has been the subject of many frontline investigations. The dominant scenarios about the origin of the Indian midocean ridge (MOR)-type mantle beneath the SCS marginal basin invoke: (i) lithospheric extension [2,10], (ii) mantle escape from below southeast Asia due to lithospheric thickening as a result of the Indo-Eurasian collision [3], and (iii) upper mantle contamination by a deep-rooted thermal plume at the fringe of the downwelling zone of Asia [11–13]. Despite a plethora of scenarios regarding the origin of the SCS mantle, the last hypothesis has gained ground owing to geophysical data predicting a ~1900 km-deep, plume-like mantle structure below the Hainan Island/Leizhou Peninsula area to the west of the SCS [14,15]. The fact that intraplate (plume-related) basalts in most areas surrounding the SCS are older than ~33 Ma implies that the Hainan diapir was active before the genesis of the SCS and, by extension, might have promoted the opening of the SCS marginal basin [16].

Herein we present a study of the petrographic and geochemical characteristics of a suite of basalt lavas we collected from the Nanyue intraplate seamount in the Southwest sub-basin of the SCS. We leverage our whole-rock compositional data with high-precision Sr–Nd–Pb–Hf–Os isotope measurements to examine whether in situ petrologic processes were capable of creating Indian MOR-type mantle reservoirs underneath the SCS. We also present $^{40}$Ar/$^{39}$Ar age results to check if basalt compositions vary as a function of age. In addition, we combined our data with those available from previous investigations to provide an integrated view of the lithospheric evolution of the Southwest sub-basin of the SCS. We suggest that the thermal perturbation imparted by the Hainan plume upon the mantle beneath the SCS gave rise to a phase of extensive postspreading magmatism during the Late Cenozoic.

## 2. Geological Background

### 2.1. Geological Setting of the South China Sea (SCS)

The SCS is a marginal sea in the western side of the Pacific Ocean (Figure 1a), encompassing an area of ~3.7 × 10$^6$ km$^2$. It stretches from NNE to SSW for ~2.1 × 10$^3$ km and has a nearly rhomboidal shape. The central part of the SCS basin has a mean water depth of ~4.3 km. In physiographic terms the SCS is bounded by South China mainland on the north, the Indochinese Peninsula on the west, Taiwan and the Philippine/Palawan archipelago on the east, and the island of Borneo on the south. Owing to its small size, ease of access, and unique location at the intersection of the Eurasia, Philippine Sea, and Indo-Australia plates, the SCS is regarded as a natural laboratory for studying global-scale plate-tectonic geodynamics.

Since the Late Paleozoic, the SCS and surrounding areas have undergone many first-order geotectonic events. The Mesozoic collision between the Eurasian plate and the India-Australian plate coupled with the northward subduction of the paleo-Pacific plate eventually dispelled Paleo-Tethys in the broader area of the present-day SCS [17]. Continental rifting followed by seafloor spreading led to the opening of the SCS in the Cenozoic era [16,18]. These geological processes collectively shaped

the architecture of the SCS area dividing it into three tectono-structural domains: (i) the northern continental margin, (ii) the southern continental margin, and (iii) the ocean basin. The northern continental margin is generally considered as a hyperextended "intermediate-type" margin. The southern continental margin is composed of a number of continental microblocks (Dangerous Grounds, Reed Bank, North Palawan, and Luconia) that amalgamated with the Borneo Block after the closure of the proto-SCS (e.g., [17] and references therein). Moreover, the ocean basin of the SCS is subdivided into three extensional sub-basins: the Northwest sub-basin, the East sub-basin, and the Southwest sub-basin (Figure 1b).

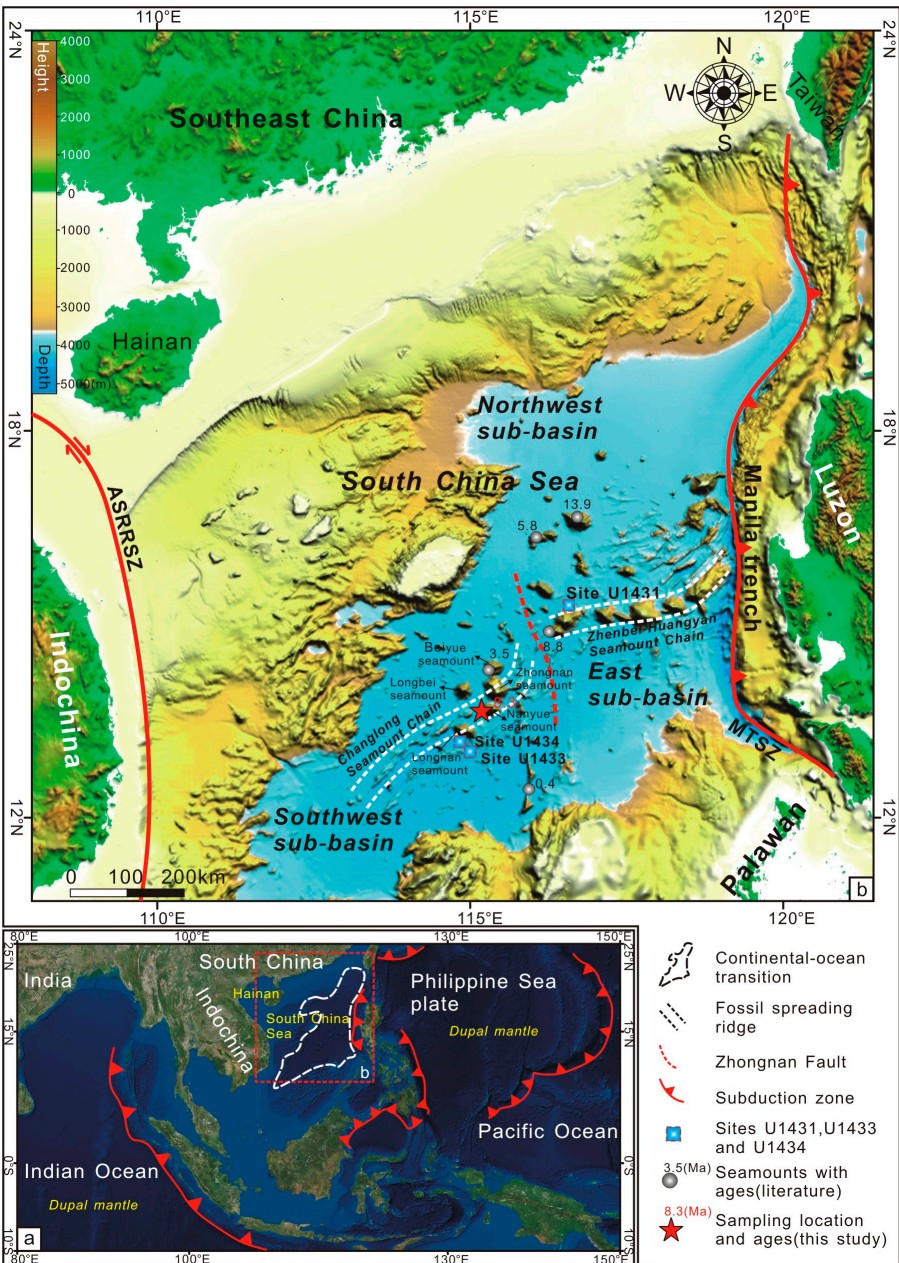

**Figure 1.** (**a**) Schematic tectonic framework of the southwestern Pacific basins showing major tectonic subdivisions (modified after [5]); (**b**) Geomorphological-bathymetric map illustrating the South China Sea (SCS) area (modified after [19]). Gray dots represent previously dated seamounts [20]. Blue squares represent the International Ocean Discovery Program (IODP) Sites U1433 and U1434 [5]. The location of the Nanyue seamount is also marked by a red asterisk. Abbreviations (in alphabetical order): ASRRSZ = Ailao Shan-Red River Shear Zone; MTSZ = Mariana Trench Subduction Zone.

Several works have suggested that a first period of NNW-SSE spreading occurred in the East and Northwest sub-basins during the Oligocene (33–26 Ma). This was followed by two episodes of NW-SE extension during the Late Oligocene (26–22 Ma) and Early to Middle Miocene (22–15.5) Ma in the East and Southwest sub-basins [19,21]. Ocean floor spreading ceased at ~15.5 Ma, leaving three fossil spreading centers in the SCS basin. Soon after the termination of seafloor spreading, extensive intraplate magmatism started affecting the broader SCS region (i.e., Indochina Block, Hainan Island, etc.) [16]. Postspreading volcanism resulted in the formation of seamount chains and alkaline lava flows in the abyssal plain and the continental slopes of the SCS, respectively [5,22]. Today, intraplate volcanism is active at the northern margin of the SCS and the Indochina block region [16].

## 2.2. The Southwest Sub-Basin

The Southwest sub-basin is on the southwest side of the central abyssal plain of the SCS (Figure 1b). It tilts from NW to SE for ~0.6 km and has an almost triangular shape, occupying an area of $1.8 \times 10^6$ km$^2$. In the Southwest sub-basin, the water depth ranges between 4.0 and 4.4 km. It is bounded by the Indochina block on the west, the Zhongnan transform fault on the east, and the Palawan–Borneo trough on the south. Drilling of the SCS seafloor during the International Ocean Discovery Program (IODP) Expedition 349 [23] has recovered oceanic crustal rocks near the fossil ridge of the Southwest sub-basin (Sites U1433 and U1434; Figure 1b). Before reaching the oceanic crust, a pelagic sedimentary cover approximately 300–850 m in thickness, composed of turbidites, unconsolidated sands to sandstones, carbonates, and siltstones, was recovered. The collected oceanic rock samples were primarily enriched (E)-MOR−type basalts [5]. Geochronological dating of these rocks using the Ar-Ar method yielded ages of ~16.3 to 17.3 Ma [24], corresponding to the stage of cessation of spreading in the Southwest sub-basin.

The seafloor of the Southwest sub-basin is relatively flat with a few topographic anomalies (i.e., sea knolls). However, a typical feature of the seabed of the Southwest sub-basin is a NE/SW-trending undersea mountain range with conspicuous volcanic seamount morphological characteristics (Figure 1b). This intermittent volcanic belt is known as the Changlong seamount chain. It includes a few tens of identified submarine volcanoes (i.e., Changlong, Nanyue, Longnan, Longbei and Zhongnan), stretching over 400 km from the northwestern end of the Spratly (/Nansha) Islands in the south to the western end of the Scarborough (/Zhenbei-Huangyan) seamount chain in the north (Figure 1b). Seamounts in the Changlong volcanic chain have heights ranging between 200 and 3100 m above the seafloor and a NE/SW-dominant elongation direction.

In the course of the IODP Expedition 349, several specimens of intraplate basalts were recovered at sites U1433 and U1434 in the Southwest sub-basin (Figure 1b) [23]. However, several parts of the fossil ridge that crosses the Southwest sub-basin were left out of sampling. This leaves a large gap in our understanding of the evolution history of the SCS. Our present study of volcanic rocks from the northern edge of the Changlong seamount chain will help us better understand the puzzling geodynamic evolution of the broader SCS region.

## 3. Sampling and Petrography

### 3.1. Sampling Strategy

Our investigation focused on the Nanyue seamount located in the northern part of the Changlong seamount chain in the Southwest sub-basin of the SCS (latitude ($\varphi$): 13°40′44″ N, longitude ($\lambda$): 115°16′22″ E; Figure 1b). Because of the anomalous topography of the seafloor of the SCS the recognition of volcanic seamounts might be a rather challenging task. For that reason, we used high-resolution, underwater topographic maps to find the exact location of the Nanyue volcanic seamount. We also took advantage of pre-existing bathymetric multibeam data to make sure that our samples were not collected from tectonically uplifted areas (i.e., anticlines) but from the flanks of the target seamount.

High-resolution bathymetric mapping revealed that the Nanyue volcanic seamount has the shape of an elliptic cone with a radius of ~22 km at the base, occupying an area of ~370 km$^2$. Samples were collected during cruise R/V Ocean No. 4 carried out by the Guangzhou Marine Geological Survey (GMGS), Ministry of Natural Resources of the People's Republic of China. Sampling was done by dredging, which represents an efficient way to recover large volumes of rocks from the oceanic environment [25]. A total of 8 rock specimens were taken from two lava breakout sites on the flanks of the Nanyue seamount at water depths between ~1020 m and ~1270 m. Each site occupied an area of a few ten square meters. Four samples were collected from each of these sites. In particular, samples HD66-1, HD66-3, HD66-7, and HD66-8 were taken from the shallowest site (~1020 m), whereas samples HD66-2, HD66-4, HD66-5, and HD66-6 were collected from the deepest site (~1270 m). However, two volcanic rock samples from the shallowest site (HD66-7 and HD66-8) were extremely altered and, thus, were excluded from further examination. Most samples were angular, indicating that they were products of in situ explosive volcanism. Sampling was undertaken to find out if postspreading magmatism in the investigated sub-basin was compositionally and isotopically heterogeneous by comparing the results of our research with those of previous studies.

*3.2. Petrography*

3.2.1. Macroscopic Description

In hand specimens, rock samples from the shallowest site exhibited a light to dark gray color, whereas those collected from the deepest site displayed a shiny black tint. All specimens showed an isotropic igneous fabric typical of mafic volcanic rocks (Figure 2a). With one exception, all samples showed an aphanitic structure. The exception, sample HD66-4, was characterized by a microporphyritic structure and was collected from the deepest sampling site. One of the most remarkable macroscopic features of all investigated samples was their porous structures represented by the high number of visible vesicles they contained (Figure 2a). These former gas-hosting bubbles were spherical to ovoid in shape and ranged in size between 1 and 5 mm. The fossil bubbles commonly occupied ~20% of the total volume of a single hand specimen, although a couple of samples with a greater volume percentage of vesicles (~30% to 35%) were studied as well. Our observations indicated that the specimens with the largest vesicles (≥3 mm) were collected at a shallow depth (~0.5 m) from the surface of the investigated seamount. This is consistent with a scenario invoking gas accumulation in the upper part of a lava flow aided by gradual coalescence of contiguous gas bubbles due to decompression. In a few hand specimens the cavities may be partially or entirely filled with pasty to yellow, unidentified secondary minerals forming amygdules (Figure 2a). A few specimens from the deepest sampling site had thin, glassy margins owing to rapid cooling. They most likely represent fragments of bulb-shaped pillow lavas.

We note that the investigated rock samples appeared generally fresh, having preserved their original volcanic features.

3.2.2. Microscopic Description

For the purpose of the present investigation we prepared three polished thin sections from each of the six hand specimens of (apparently fresh) volcanic rocks we collected from the Nanyue seamount. All thin sections were studied under transmitted light, using a Nikon Eclipse LV100 POL petrographic microscope at the School of Marine Sciences, Sun Yat-sen University (SMS-SYSU), Guangzhou, China. The mineral modes of the investigated rocks were determined by visual estimation methods.

The polished thin sections of the investigated rock samples showed a porphyritic texture with a small amount of phenocrysts (~15% to 25% modal) set in a dark amorphous mesostasis (~75% to 85% modal) composed mainly of volcanic glass that locally contained microlites of plagioclase (≤100 μm), clinopyroxene (≤80 μm), and opaque metallic minerals (≤30 μm; Figure 2b). We note that we did

not observe any significant mineralogical or microtextural differences among the studied samples. Therefore, in this chapter we will describe them as a single group of mafic rocks.

Modal analyses of the phenocrysts gave averages of ~50% to 60% plagioclase (anorthite), ~40% to 50% clinopyroxene (titanoaugite), and trace amounts of metallic minerals (i.e., Fe–Ti oxides), leading to the classification of the investigated mafic volcanic rocks as basalts. Plagioclase formed large (0.5–1.0 mm), hypidiomorphic to idiomorphic crystals (Figure 2b). Plagioclase grains sporadically displayed discontinuous normal zoning due to a nongradual decrease in the CaO content from their cores to the rims (Figure 2b). Clinopyroxene phenocrysts were generally smaller than 0.6–0.7 mm (Figure 2c). They were subhedral to euhedral in shape, occasionally displaying weak pleochroism, undulatory extinction, and partially resorbed grain boundaries (Figure 2c). Clinopyroxene crystals may be converted to a mixture of clay minerals and oxyhydroxides (Figure 2d), especially along grain boundaries and brittle fractures. Opaque minerals were represented by ilmenite and magnetite. The size of these metallic minerals did not exceed 50 μm. They commonly occurred as globular to subhedral grains poikilitically enclosed in clinopyroxenes and less commonly as subhedral to euhedral microcrysts scattered within the glassy matrix of the investigated basalts.

The matrix was generally unaltered, but in places it may become partially devitrified. In a few of the investigated polished, thin sections, the vesicles were filled with secondary calcite and rarely with Ca-rich zeolites.

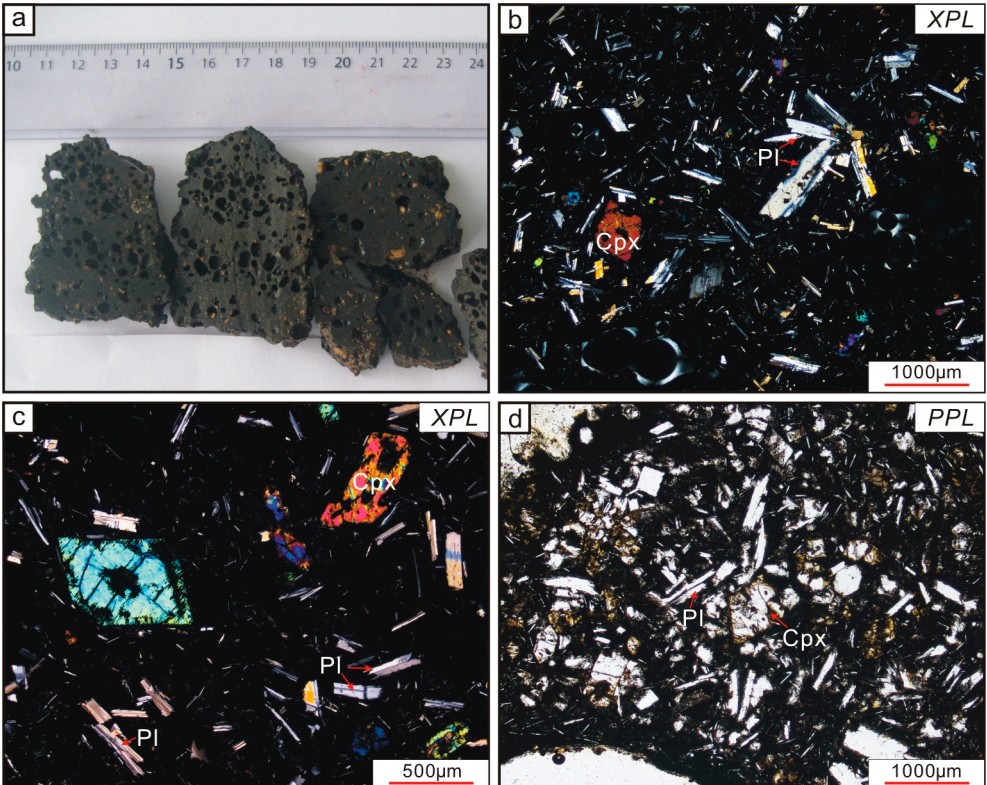

**Figure 2.** (**a**) Macroscopic photo of various rock specimens collected from the Nanyue seamount showing a pervasive isotropic, aphanitic fabric, a porous structure, and some amygdules. Photomicrographs of basalts from the Nanyue seamount showing: (**b**) porphyritic texture; (**c**) Clinopyroxene crystals with partially resorbed grain boundaries; and (**d**) Clinopyroxene crystals marginally converted to a mixture of clay minerals and oxyhydroxides. With one exception (photomicrograph d), all photomicrographs were taken under cross-polarized (transmitted) light (XPL). Photomicrograph d was taken under plane-polarized (transmitted) light (PPL). Abbreviations (in alphabetical order): Cpx—clinopyroxene; Pl—plagioclase.

## 4. Methods and Results

### 4.1. Analytical Methods

Four out of six unaltered basalt samples were selected from the Nanyue seamount for bulk-rock analyses of major element oxides and trace elements. We actually analyzed two basalt specimens from each sampling site on the flanks of the Nanyue seamount. Basalt samples from the deepest site of the investigated seamount shared quite similar mineralogical and microtextural characteristics. Therefore, the selection of samples for analytical purposes was based on their structural differences (aphanitic (HD66-2) vs. microporphyritic structure (HD66-4)). Major element oxide analyses were conducted on fused glasses using X-ray fluorescence (XRF) at the Guangzhou Institute of Geochemistry, Chinese Academy of Sciences (GIG-CAS) (Guangzhou, China). Trace element analyses were performed employing an inductively coupled plasma-mass spectrometer (ICP-MS) Thermo X Series II at the Radiogenic Isotope Facility, School of Earth and Environmental Sciences, University of Queensland (RIF-SEES-UQ), Brisbane, Australia. Three basalts samples (HD66-1, HD66-2, and HD66-3) were analyzed for their Sr–Nd–Pb–Hf isotopes. Strontium–Nd–Pb isotope ratios of bulk-rock samples were determined using a multicollector (MC)-ICP-MS (Nu Plasma HR) at the RIF-SEES-UQ. Hafnium isotope analyses were carried out using a Micromass IsoProbe MC-ICP-MS at the GIG-CAS. Furthermore, two basalt samples (HD66-1 and HD66-2) were analyzed for their Os isotopes. Osmium concentration and isotope analyses were carried out employing a Thermo Fisher Triton thermal ionization mass spectrometer (TIMS) at the GIG-CAS. In addition, two groundmass samples (HD66-1 and HD66-2) were dated by the $^{40}$Ar/$^{39}$Ar incremental heating method using the TRIGA CLICIT nuclear reactor at Oregon State University (OSU), Corvallis, USA. Details of all the analytical procedures are provided in the Supplementary Materials file. Geochemical and isotopic data are given in Table S1 in the Supplementary Materials file. Geochronological data are presented in Table S2 in the Supplementary Materials file.

### 4.2. Results

#### 4.2.1. Major and Trace Element Analyses

Basalts from the Nanyue seamount showed limited variations in their concentrations of major-element oxides ($SiO_2$ = 47.89–48.80 wt %, $Al_2O_3$ = 16.40–16.73 wt %, CaO = 8.56–8.74 wt %, MgO = 5.61–6.05 wt %, $Na_2O$ = 2.90–3.34 wt %, $K_2O$ = 2.04–2.13 wt %, $TiO_2$ = 2.75–2.81 wt %, and $P_2O_5$ = 0.63–0.87 wt %). They had high $Na_2O/K_2O$ ratios (1.36–1.64). On the total alkalis ($Na_2O + K_2O$) vs. $SiO_2$ (TAS) diagram [26] their compositions plot in the alkaline series field (Figure 3), indicating that they comprised a geochemically homogeneous group of rocks. This was corroborated by their almost equal magnesium numbers ($100 \times Mg^{2+}/(Mg^{2+}+Fe^{2+})$; $Mg^{\#}$ = 57–59). The Nanyue basalts had low loss on ignition (LOI) values (1.51–1.82 wt %) consistent with the absence of alteration-related minerals.

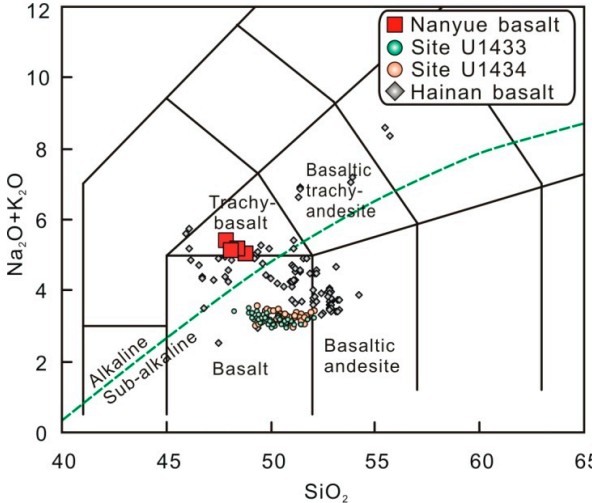

**Figure 3.** $SiO_2$ vs. total alkalis ($Na_2O + K_2O$) diagram of [26]. The alkaline–subalkaline boundary is after [27]. Compositional data for volcanic rocks from the Hainan Island and the IODP Sites U1433 and U1434 are from [5,13].

When plotted on primitive mantle (PM)-normalized diagrams, the investigated basalts exhibited enrichments in specific high-field-strength elements (HFSE) such as U, Ta, and Zr (Figure 4a). They also displayed weak, positive anomalies in some large-ion lithophile elements (LILE), namely, Ba and Sr (Figure 4a). Furthermore, they showed mild, positive Ho spikes and negative Th, Nb, and Y anomalies (Figure 4a). The Nanyue basalts showed smoothly descending chondrite-normalized REE patterns from the light (L)REE to the heavy (H)REE (i.e., La/Sm = 5.64–5.96 versus Sm/Yb = 3.55–3.63; Figure 4b). In addition, they displayed chondrite-normalized REE profiles characterized by weak, negative Sm and Yb anomalies (Figure 4b).

In general, the right-sloping "topography" of the PM-normalized trace element profiles of the Nanyue basalts follows that of the PM-normalized diagrams of a number of basalts from the adjacent Hainan Island [11,12,28] (Figure 4a). Compared to the E-MOR–type basalts collected from the IODP Sites U1433 and U1434, the investigated basalts are slightly depleted in the HREE and enriched in the MREE and LREE (Figure 4b). Collectively, our geochemical data show that the compositional signatures of basalts from the Nanyue seamount are analogous to those of typical OIB-type rocks [29] (Figure 4).

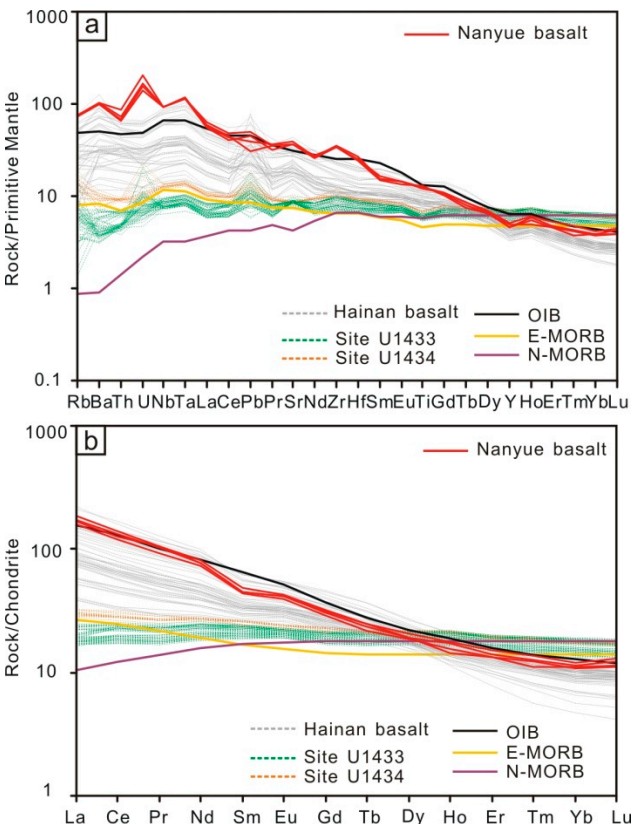

**Figure 4.** (**a**) Primitive mantle (PM)-normalized multielement profiles and (**b**) chondrite-normalized lanthanide profiles for the Nanyue basalts. The PM- and chondrite-normalizing data are from [29]. Sources of data for reference normal (N)-MORB, E-MORB, and (OIB) are from [29]. Compositional data of volcanic rocks from the Hainan Island and the IODP Sites U1433 and U1434 are from [5,13].

### 4.2.2. Sr–Nd–Hf–Pb Isotope Systematics

Basalts from the Nanyue intraplate seamount had quite uniform $^{87}Sr/^{86}Sr$ (0.704222–0.704325) and $^{143}Nd/^{144}Nd$ (0.512839–0.512881) ratios. In addition, they showed a limited range of $\varepsilon Nd(t)$ values between +4.0 and +4.8. In the $\varepsilon Nd(t)$ vs. $^{87}Sr/^{86}Sr$ diagram, the isotopic compositions of the investigated basalts plot in the field of the Indian Ocean MORB [30] and in the overlapping area between the fields of the Indian Ocean MORB [30] and the Hainan Island OIB [11,12,28,31] (Figure 5a).

The Nanyue basalts had positive $\varepsilon Hf(t)$ values (4.7–5.0). In the $\varepsilon Hf(t)$ vs. $\varepsilon Nd(t)$ diagram, the compositions of the investigated basalts plot in the bottom-left part of the field of the Indian Ocean MORB; they were almost similar to the average composition of OIB from the Hainan Island [11,12,28,31] (Figure 5b). From this diagram it becomes obvious that there was no decoupling of $\varepsilon Nd$ from $\varepsilon Hf$ in the mantle source of the Nanyue basalts.

The $^{206}Pb/^{204}Pb$ ratio of the investigated basalts varied between 17.73 and 18.26. These $^{206}Pb/^{204}Pb$ ratios were less radiogenic than those of basalts from the East sub-basin of the SCS (18.38–18.56) [5] and basaltic breccia clasts from the Daimao seamount in the Northwest sub-basin (18.67–18.97) [32]. However, the $^{206}Pb/^{204}Pb$ ratio of the Nanyue basalts were closer to those of oceanic basalts from the central Southwest Indian ridge (39–41°E) where modern MORB with a very low $^{206}Pb/^{204}Pb$ ratio (16.6–17.5) occurred [33]. In the $^{207}Pb/^{204}Pb$ vs. $^{87}Sr/^{86}Sr$ and $^{206}Pb/^{204}Pb$ vs. $\varepsilon Nd(t)$ diagrams, the Nanyue basalts showed analogous isotopic compositions to those of the Indian Ocean MORB [30] (Figure 5c,d).

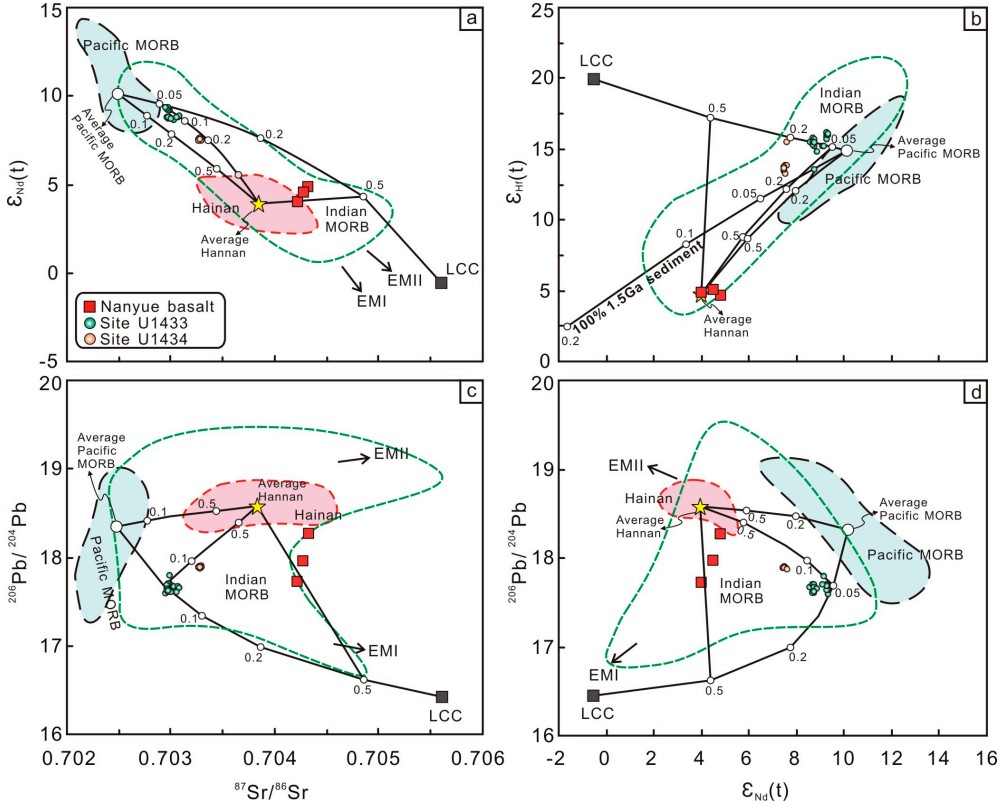

**Figure 5.** Isotopic plots of: (**a**) εNd(*t*) versus ⁸⁷Sr/⁸⁶Sr, (**b**) εHf(*t*) versus εNd(*t*), (**c**) ²⁰⁶Pb/²⁰⁴Pb vs. ⁸⁷Sr/⁸⁶Sr, and (**d**) ²⁰⁶Pb/²⁰⁴Pb versus εNd(*t*) for the Nanyue basalts. We note that εNd and εHf were normalized to the ¹⁴³Nd/¹⁴⁴Nd and ¹⁷⁶Hf/¹⁷⁷Hf ratios of the chondrite uniform reservoir (CHUR; 0.512630 and 0.282785, respectively) [34]. Sources of data for comparison: the Pacific MORB and the Indian MORB are from [30] and the Hainan OIB is from [11,12,28,31]. Sources of data used for the end-member mixing calculations: the average isotopic composition of the Hainan OIB is from [11,12,28,31], the average isotopic composition of the Pacific MORB is from [30], and the isotopic composition of the lower continental crust (LCC) is from [35]. Binary lines corresponding to the isotopic compositions resulting from various mixing models are from [5]. Each white dot on the mixing lines represents various increments of melt fractions from the average Hainan OIB, the average Pacific MORB, and the LCC. Data references of enriched mantle (EM)-I and -II end-members are from [36]. Compositional data of volcanic rocks from the IODP Sites U1433 and U1434 are from [5].

Furthermore, in the ²⁰⁸Pb/²⁰⁴Pb vs. ²⁰⁶Pb/²⁰⁴Pb diagram, the isotopic compositions of the Nanyue basalts plot above the Northern Hemisphere reference line (NHRL) [37], indicating that they were derived from an Indian-type mantle source with a Dupal anomaly (Figure 6a) [36]. In the ²⁰⁷Pb/²⁰⁴Pb vs. ²⁰⁶Pb/²⁰⁴Pb plot, the isotopic compositions of the Nanyue basalts were well bracketed by the 4.48 Ga and 4.568 Ga geochrones (Figure 6b), implying the possible involvement of an old depleted mantle (DM) source [38] in their formation.

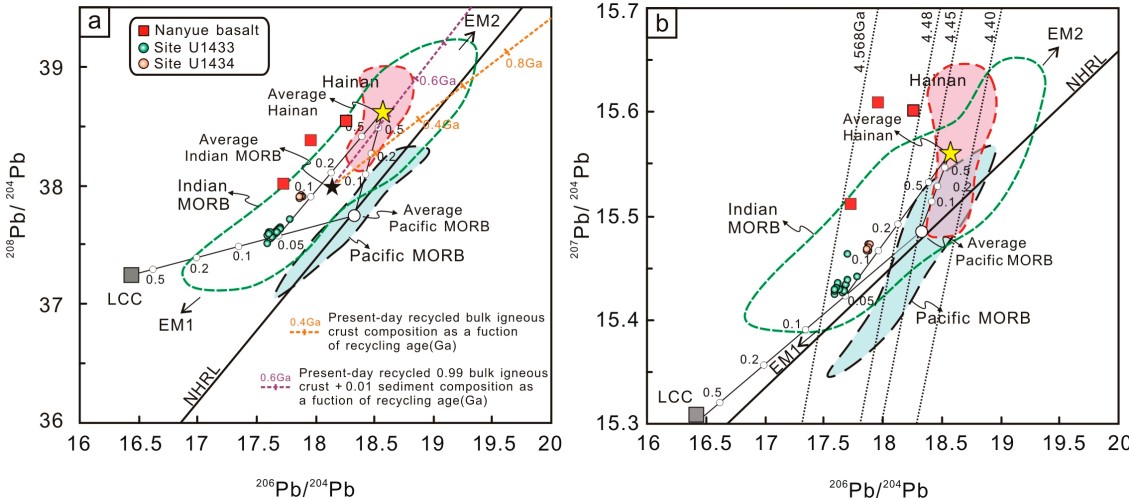

**Figure 6.** Isotopic plots of: (**a**) $^{208}Pb/^{204}Pb$ vs. $^{206}Pb/^{204}Pb$ and (**b**) $^{207}Pb/^{204}Pb$ vs. $^{206}Pb/^{204}Pb$ for the Nanyue basalts. Sources of data for comparison: the Pacific MORB and the Indian MORB are from [30] and the Hainan OIB is from [11,12,28,31]. The Northern Hemisphere reference line (NHRL) is from [37]. Sources of data used for the end-member mixing calculations: the average isotopic composition of the Hainan OIB is from [11,12,28,31], the average isotopic composition of the Pacific MORB is from [30], and the isotopic composition of the LCC is from [35]. Each white dot on the mixing lines represents various increments of melt fractions from the average Hainan OIB, the average Pacific MORB, and the LCC. Data references of enriched mantle (EM)-I and -II end-members are from [36]. The geochrones in (a) are from Figure 3(a) in [13]. Compositional data of volcanic rocks from the IODP Sites U1433 and U1434 are from [5].

### 4.2.3. Re–Os Isotope Systematics

The Nanyue basalts had low Os content (60.37–85.13 ppt) and moderate Re concentrations (249.60–255.11 ppt). In the Re vs. Os diagram, the investigated basalts plot in the field of OIB and high-MgO lavas (Figure 7a). In addition, the Nanyue basalts were poorer in Re than most MORB-type rocks [39] but contained more Re than the Cenozoic OIB lavas from the near Hainan Island [13]. They also plot outside of the compositional field of plume-related picrites from Iceland (Baffin Island) and west Greenland (BIWG) [40–45].

The Nanyue basalts had $^{187}Re/^{188}Os$ ratios ranging between 14.27 and 20.50 and $^{187}Os/^{188}Os$ ratios varying from ~0.19 to ~0.21. In the $^{187}Os/^{188}Os$ vs. Os logarithmic diagram, the investigated basalts plot out of the field of abyssal peridotites (AP) from all over the globe [39,40] (Figure 7b). Furthermore, they had much higher $^{187}Os/^{188}Os$ ratios compared to that of the present-day primitive upper mantle (PUM; $^{187}Os/^{188}Os = 0.1296 \pm 0.0008$; Figure 7b) [46].

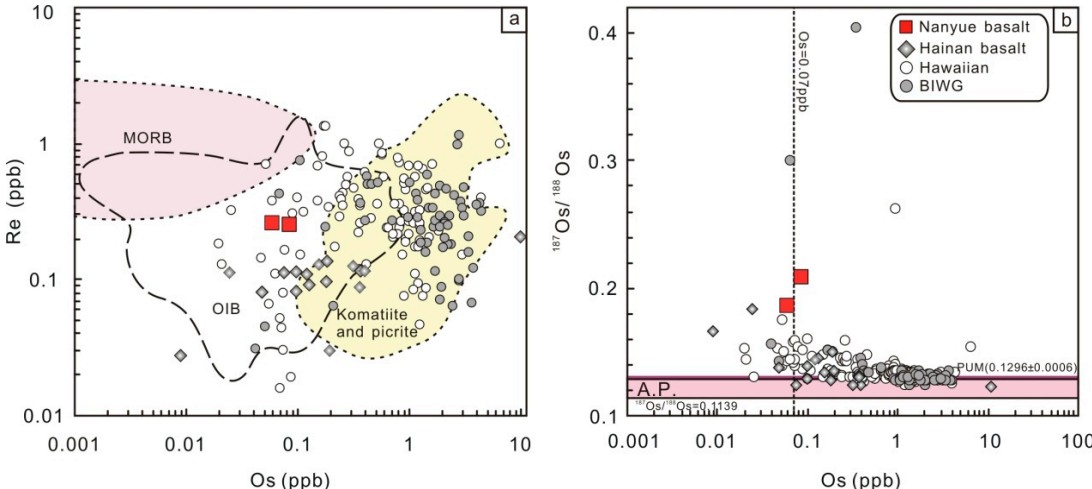

**Figure 7.** (**a**) Re concentrations vs. Os concentrations and (**b**) $^{187}$Os/$^{188}$Os vs. Os contents for the Nanyue basalts and comparison with those in the Hainan basalts [13], the plume-related lavas from Hawaii [41–43], and the early (ca. 60 Ma) plume-related picrites from Iceland (Baffin Island) and West Greenland (BIWG) [40,44,45]. The $^{187}$Os/$^{188}$Os ratio (0.1296 ± 0.0008) for the present-day primitive upper mantle (PUM) in (b) is from [46]. The $^{187}$Os/$^{188}$Os ratio in (b) for the Hawaiian and the Iceland plume-related lavas have been corrected to their initial values. Data for abyssal peridotites (AP) in (b) are from [39,40].

### 4.2.4. $^{40}$Ar/$^{39}$Ar Geochronological Dating

Basalt sample HD66-1 from the shallowest sampling site on the flanks of the Nanyue seamount (1022 m depth) gave a plateau age of 8.26 ± 0.06 Ma with an inverse isochron age of 8.15 ± 0.06 Ma (mean square weighted deviation (MSWD) = 4.01; Figure 8a). Moreover, basalt sample HD66-2 from the deepest sampling site (1273 m depth) gave a plateau age of 8.29 ± 0.06 Ma with an inverse isochron age of 8.17 ± 0.06 Ma (MSWD = 4.27; Figure 8b). This indicates that basalt lava from the deepest parts of the Nanyue volcanic seamount is older than that from the shallowest parts of the volcano.

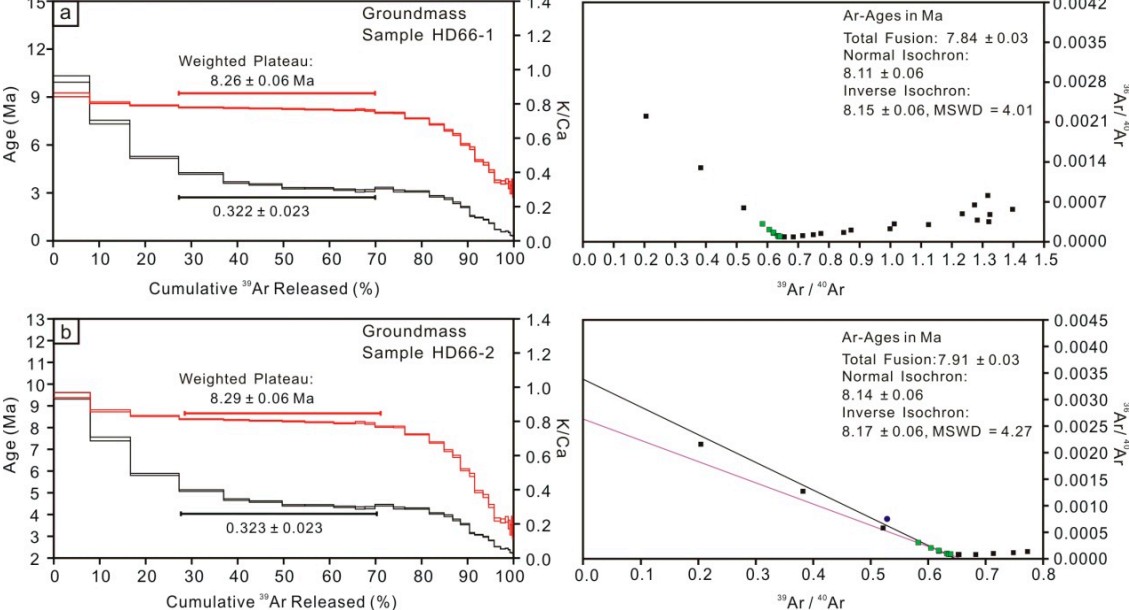

**Figure 8.** (**a**,**b**) Age spectra, integrated and plateau ages, and isochron diagrams of the Nanyue basalts.

## 5. Discussion

*5.1. Potential Effects of Melt Fractionation, Crustal Contamination, and Post-Solidification Alteration*

In order to probe the nature of the mantle source domain(s) that contributed to post-spreading magmatism in the Southwest sub-basin of the SCS, we will first explore the impacts of melt fractionation, crustal contamination, and postmagmatic alteration on the compositions of the Nanyue basalts. This will help us identify the petrologic conditions under which magma emplacement occurred and enable further assessment of the geochemical and isotopic compositions of the source materials that were involved in the genesis of the investigated rocks.

The investigated rocks are small-volume, alkaline basalts characterized by low concentrations of MgO (5.61–6.05 wt %), Ni (105–154 ppm), and Cr (145–162 ppm), and moderate Mg# values (57–59; see Table S1 in the Supplementary Materials file). These geochemical signatures do not match those of primitive magmas, implying that the parental melts of the Nanyue basalts must have undergone fractional crystallization to some extent en route to shallow lithospheric levels and/or during their storage in a magma chambers below the investigated seamount.

The ascending magmas passed through the crust before erupting on the seafloor. This opens the possibility of crustal contamination. Indeed, the Nanyue basalts have Nb/Ta ratios (13.55–14.01) that range between those of the PM (17.5 ± 0.5) and the continental crust (~11 to 12) [47]. This compositional signature is most likely attributed to contamination of the parental melts of the investigated rocks with crustal components [13]. However, it is not clear how crustal contamination happened, as several mechanisms have been proposed in the literature to explain crustal inputs in presumably mantle-derived magmas (i.e., slab subduction, mantle melt–crust interaction, and crustal delamination followed by erosion) [4,5,12,13].

The $^{187}Os/^{188}Os$ ratio of a mantle melt is an extremely sensitive parameter to addition of radiogenic crustal components. Basalt samples from the Nanyue seamount have low Os concentrations and radiogenic $^{187}Os/^{188}Os$ ratios. This is often interpreted as a typical "hallmark" of crustal contamination (e.g., [48]). However, our Os-isotopic data are not in favor of contamination of the parental magmas of the Nanyue basalts by crustal materials. Although two basalt analyses constitute a dataset too small to be statistically robust, their $^{187}Os/^{188}Os$ ratios decreased with decreasing Os concentrations. For instance, the basalt sample (HD66-2) with the lowest Os concentration (60.37 ppt) and the highest $^{187}Os/^{188}Os$ ratio (0.1856) had the lowest $^{87}Sr/^{86}Sr$ ratio (0.704222) and $\varepsilon$Nd (4.0) of all the basalt samples analyzed, which would not be the case if crustal contamination was responsible for its Os-isotopic composition. In addition, the lack of a co-variation between the Os content and other trace element contents or radiogenic isotopes implies that crustal assimilation did not significantly affect the Os-isotopic variation of the Nanyue basalts. Therefore, we conclude that the Os-isotopic compositions of the Nanyue basalts are most likely the original ones.

We note that the Nanyue basalts have been affected by low levels of postmagmatic alteration. For that reason, postsolidification alteration alone is not likely to be the cause of their distinctive geochemical signatures, particularly for alteration-resistant ratios such as Nd/Pb, La/Nb, and Nb/Th.

We note that previous studies have shown that Late Cenozoic magmas from the SCS and surrounding regions (e.g., Hainan, Indochina) underwent fractional crystallization but were not significantly affected by crustal contamination during their ascent to the surface (e.g., [49] and references therein). In contrast, our findings indicate that the parental magmas of the Nanyue basalts have been affected by crustal contamination to some extent. Assimilation of crustal materials is capable of changing the isotopic signatures of the basalt-forming melts. Therefore, the Sr–Nd–Pb–Hf isotope ratios of the Nanyue basalts must not be analogous to those of their parental magmas. Consequently, they cannot help us identify the thermal conditions under which mantle melting occurred and enable further examination of the depth and degree of mantle melting. Nevertheless, the isotopic systematics of the Nanyue basalts offer a unique opportunity to study the effect of crustal contamination on the genesis of postspreading volcanic rocks with a Dupal/Indian-type isotopic anomaly.

### 5.2. Genesis of the Dupal Isotopic Anomaly in the Mantle under the Southwest Sub-Basin

Clearly, basalts from the Nanyue seamount have lower $^{206}$Pb/$^{204}$Pb ratios and significantly higher $^{87}$Sr/$^{86}$Sr ratios compared to those of the average Pacific MORB (Figure 5c). Therefore, melting of a Pacific-type mantle reservoir alone could not sufficiently explain the isotopic signatures of the investigated basalts. A plethora of petrologic studies have indicated that mixing of (at least) two mantle components is essential for the genesis of a source region with a Dupal isotopic anomaly [5,18]. Indeed, the isotopic variations of the Nanyue basalts could be explained by mixing between a depleted MORB-type mantle (DMM) source and enriched mantle (EM) components [36]. Consequently, the question arises as to what the origin of the inferred EM components was.

Could it be recycled pelagic sediments of Mesoproterozoic age (~1.5 Ga)? Recycling of ancient pelagic sediments is capable of causing limited decoupling of $\varepsilon$Hf from $\varepsilon$Nd because the Nd/Hf ratio of ocean sediments is 1.5 times greater than that of the DMM ([5] and references therein). On the $\varepsilon$Hf($t$) vs. $\varepsilon$Nd($t$) diagram (Figure 5b) it becomes evident that the Hf–Nd isotopic compositions of the Nanyue basalts cannot be explained by recycling of old marine sediments. Furthermore, melting of terrigenous sediments derived from continental regions could not be the source of the inferred EM components in the mantle underneath the SCS (Figure 5c,d). This is because melting of terrigenous sediments in the lid of a subducted lithospheric slab cannot account for the relatively low $^{206}$Pb/$^{204}$Pb ratio of the Nanyue basalts [5].

Could another possible origin for the EM component discovered in the Nanyue basalts be the subcontinental lithospheric mantle (SCLM)? The SCS is a marginal basin formed after break-up of the southeastern part of Eurasia. Therefore, the question arises as if partial melting of the (metasomatized) SCLM that once occupied the region where the present-day SCS is located could explain the relatively enriched Sr–Nd and depleted Pb isotopic compositions of the Nanyue basalts (Figure 5). It is known that melting of the SCLM is capable of causing significant Nd–Hf isotopic decoupling in the resultant magmas [4]. However, this is not the case for the investigated basalts (Figure 5b). Furthermore, the SCLM mantle is generally considered to be depleted [5]. Therefore, melting of the SCLM cannot explain the enriched trace element patterns of the Nanyue basalts. In addition, the investigated rocks have PM-normalized diagrams showing positive Ta anomalies (Figure 4a) unlike basalts derived from melting of a SCLM source [49] and references therein, [50,51]. Moreover, the investigated basalts have more radiogenic $^{187}$Os/$^{188}$Os ratios (~0.19 to 0.21) than that inferred for the SCLM (~0.15) [52,53].

An alternative scenario to interpret the EM-like isotopic signature in the upper mantle source of the Nanyue basalts invokes recycling of petrologic materials derived from delamination of the lower continental crust (LCC). However, a process of thermo-chemical erosion of the LCC alone would most likely give rise to the genesis of melts with much higher $\varepsilon$Hf($t$) values than those of the investigated rocks (Figure 5b).

Another hypothesis to explain the EM-like isotopic signatures in the Nanyue basalts highlights the role of the adjacent Hainan mantle plume. This is because the Os concentrations and $^{187}$Os/$^{188}$Os ratios of the investigated basalts are analogous to those of the Hainan OIB (Figure 7). However, the Sr–Nd–Pb–Hf isotopic compositions of the Nanyue basalts do not always plot between those of the average Hainan OIB and the average Pacific MORB (Figure 5a,c,d).

To explain the genesis of the EM-like isotopic signature in the postspreading basalts of the Nanyue seamount, we applied a semiquantitative, ternary mixing model first proposed by [5]. We used this model to quantify the impact of a set of distinct petrologic components on the parental melts of the investigated rocks (Figure 5). Our plots indicate that the isotopically "DM end-member" signature of the Nanyue basalts may be produced by blending of a granulitic LCC with the average Pacific N-MORB in equal proportions (Figure 5). Then, we assessed the addition of a Hainan plume component to the resultant "DM end-member". It seems that effective mixing of (most likely less than 50%) EM-type components derived from the Hainan plume (through a buoyancy-driven melting mechanism) [54] with the "DM end-member" can reproduce the isotopic trends of the Nanyue basalts (Figure 5a,c,d). However, it seems that this model cannot sufficiently explain the relatively low $\varepsilon$Hf values of the

investigated basalts. We interpret this inconsistency an indication of a magma mixing processes. We suggest that the impact of the "DM end-member" on the Hf-isotopic signature of the Nanyue basalts was soon overprinted by large-scale mixing of the "DM end-member" with the OIB-type components derived from the adjacent Hainan diapir. This suggestion is evidenced by: (i) the fact that the upper mantle source of postspreading basalts from the SCS has been compositionally influenced by plume-type melts [4,5], (ii) the zoning texture preserved in the plagioclase crystals contained in the Nanyue basalts (Figure 2b), and (iii) the existence of clinopyroxenes with partially resorbed grain boundaries (Figure 2c).

Our two-step mixing model implies that the average Pacific N-MORB mixed with the LCC components prior to mixing with the Hainan mantle plume components. This argument necessitates that interaction of the DMM with the LCC occurred most likely within the lithospheric part of the upper mantle. This process was followed by interaction of the resultant suboceanic lithospheric mantle (SOLM) with asthenospheric plume-type components derived from the nearby Hainan mantle diapir. We note that the isotopic signatures of the Nanyue basalts provide evidence for erosion of the LCC because there is a major thermal perturbation in the upper mantle beneath the Southwest sub-basin of the SCS. It is likely that the Hainan plume was the cause of the inferred thermal anomaly in the SCS mantle.

On the basis of a similar modeling of the isotopic compositions of postspreading basalts from the Southwest sub-basin by [5], it was concluded that their isotopic trends could be reproduced by: (i) adding 5–10% LCC to the average Pacific N-MORB followed by (ii) mixing of the ensuing "DM end-member" with 5–10% of the average Hainan OIB (Figure 5). The basalts studied by [5] were collected from the IODP Sites U1433 and U1434 (Figure 1b). Both sites are located to the south of the Nanyue seamount. If we combine our results with those of [5], the proportion of plume-type components in the isotopic compositions of basalts generally increases from the south to the north of the Southwest sub-basin. If this is correct, it could be envisaged that: (i) the upper mantle under the investigated sub-basin is heterogeneous, and (ii) the influence of the Hainan diapir on the geochemical and isotopic composition of the SOLM was greater in the central parts of the SCS.

*5.3. Petrotectonic and Geochronological Implications*

The Dupal isotopic anomaly is widely regarded as a typical feature of the upper mantle below an extensive area in the Southern Hemisphere of our planet [7–9]. Therefore, a challenging question about the origin of the Dupal anomaly in the mantle below the Southwest sub-basin concerns if it represents an artifact of in situ petrologic processes or an isotopic signature that was inherited from a nonindigenous mantle source. In the latter case, the source could be a northward moving mantle from the Southern Hemisphere or even eastward flowing mantle below the southeastern China/Indochinese peninsula.

The SCS is geotectonically "locked" between three major lithospheric plates (i.e., Eurasia, Indo-Australia, and Philippine Sea plates). Moreover, the Sunda–Java volcanic arc at the south end of the SCS represents an active convergent boundary between the Eurasian plate and the Indo-Australia plate, inhibiting asthenospheric flow to the north since the Early Triassic [18]. Therefore, a Southern Hemispheric origin is not likely for the Dupal isotopic anomaly in the upper mantle below the SCS.

Paleomagnetic evidence suggests southeastern China was once part of the Gondwana supercontinent and drifted northward during the Late Paleozoic [17 and references therein]. Therefore, the question arises as to if this process induced the Dupal-like isotopic anomaly in the mantle underlying the SCS. Could this happen during the extrusion of Indochina? The truth is that the timing the extrusion occurred cannot explain the continental rifting in the SCS [55], and, thus, it is not likely that the mantle under the SCS is extruded mantle from below Indochina.

The Nanyue basalts have positive $\varepsilon$Nd($t$) values and are rich in LREE (Figure 4b). However, their Nd-isotopic compositions imply that they were most likely derived from an old and LREE-depleted mantle source [1]. This inconsistency can be reconciled by the long-term existence of a LREE-depleted

mantle source that was only recently enriched by LREE-bearing components. This is further corroborated by the results of our mixing model indicating that the Dupal isotopic anomaly was not an indigenous feature of the mantle underlying the Southwest sub-basin but a result of extensive mantle reworking [5,13,49].

Another key question regarding the origin of mantle with the Dupal isotopic anomaly beneath the Southwest sub-basin of the SCS concerns the way and time the original DM was contaminated with EM components.

Thermochronological data demonstrate that the east branch of the Hainan diapir rose up to the bottom of the continental lithosphere of southeastern Eurasia prior to the opening of the SCS (45–33 Ma) [20]. This implies that plume-type components linked to the Hainan diapir had been accumulating under the southeastern Eurasia region for a quite long period of time [34]. It is likely that this process provided the essential thermo-mechanical impetus for the initiation of continental rifting, LCC erosion, and later ocean floor spreading in the SCS area [5,12]. However, there is no indication of intraplate volcanism in the SCS throughout the opening of the marginal basin (33–16 Ma) [20]. After the cessation of spreading, the broader SCS region eventually entered a phase of intraplate magmatism. This is also supported by our geochronological data indicating that the Nanyue basalts were formed during the Tortonian. Postspreading (intraplate) volcanism in the SCS was facilitated by an eastward migration of OIB-type melt components derived from the nearby and relatively stationary Hainan plume along sloping, rheological boundary layers [23]. These components ponded in a compositionally modified uppermost mantle region and were progressively sampled by the spreading centers in the SCS. This model seems to provide a reasonable explanation for the significant role of the Hainan plume in promoting the opening of the SCS and the genesis of the Indian-type mantle under some western Pacific marginal basins of the Northern Hemisphere.

Combined data show that the age of the SCS seamounts does not become younger from the Northwest to the Southwest sub-basin, as it had been proposed by previous studies [32]. Our $^{40}$Ar/$^{39}$Ar geochronological data indicate that the Nanyue basalts are of Tortonian age (~8.3 Ma) and have alkaline geochemical affinity. We also note that previous studies have demonstrated that alkali basalts from the IODP Sites U1434 and U1433 yielded Burdigalian ages (~17.3 Ma and ~16.3 Ma) [24], whereas tholeiitic intraplate basalts from the same locations yielded Tortonian to Piacenzian ages (11.5–3.4 Ma) [28]. Our geochronological data coupled with those of earlier investigations indicate that there is no direct relationship between the geochemistry and age of postspreading volcanism in the Southwest sub-basin of the SCS.

## 6. Conclusions

The present study has led to the following conclusions:

1. Basalts from the Nanyue seamount in the Southwest sub-basin of the SCS have OIB-type geochemical affinities.
2. $^{40}$Ar/$^{39}$Ar dating indicates that these basalts were formed in the Tortonian (~8.3 Ma) and represent products of postspreading volcanism in the SCS.
3. The Sr–Nd–Pb–Hf isotopic compositions of the Nanyue basalts indicate that they were derived from an upper mantle source with a Dupal-like isotopic anomaly.
4. The inferred upper mantle source was an artifact of interaction between a Pacific-type mantle reservoir and melt components derived from thermo-mechanical erosion of the LCC and the adjacent Hainan diapir.
5. This study highlights the role of plumes in the formation of upper mantle domains with "Southern Hemispheric" isotopic anomalies under some Northern Hemispheric basins of the western Pacific.

**Supplementary Materials:** The following are available online at http://www.mdpi.com/2075-163X/9/6/378/s1, Analytical methods: Details of analytical procedures for major and trace element and isotope analyses, and geochronological dating; Table S1: Results of whole-rock and isotope analyses of the Nanyue basalts;

Table S2: Results ${}^{40}$Ar/${}^{39}$Ar geochronological dating of the Nanyue basalts. References [56–64] are cited in the supplementary materials.

**Author Contributions:** Conceptualization, H.Z., L.-F.Z., and A.K.; Formal Analysis, H.Z. and L.-F.Z.; Investigation, H.Z., L.-F.Z., A.K., G.-Q.C., and Z.-F.W.; Data Curation, H.Z. and L.-F.Z.; Writing—Original Draft Preparation, H.Z., L.-F.Z., and A.K.; Writing—Review and Editing, H.Z., L.-F.Z., A.K., G.-Q.C., Z.-F.W., and B.X.; Visualization, H.Z., L.-F.Z., A.K., and Z.-F.W.; Funding Acquisition, Z.-F.W. and B.X.

**Funding:** This work was financially supported by the National Natural Science Foundation of China (NNSFC; grant Nos. 41706055, 41776056), the Natural Science Foundation of Guangdong Province (grant No. 2018B030311030), and the National Key R & D Program of China (grant No. 2018YFC0310000).

**Acknowledgments:** We would like to express our sincere gratitude to Jian-xin Zhao and Yue-xing Feng from the Radiogenic Isotope Facility, School of Earth and Environmental Sciences, University of Queensland (RIF-SEES-UQ), Brisbane (Australia) for their assistance at the stage of the bulk-rock and isotopic analyses of our samples. The valuable comments and suggestions of two anonymous reviewers also helped us greatly improve our paper. Our thanks also go to the crew of the R/V Ocean No. 4 cruise carried out by the Guangzhou Marine Geological Survey (GMGS), Ministry of Natural Resources of the People's Republic of China.

**Conflicts of Interest:** The authors declare no conflict of interest.

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
