# Peer review of "Post-spreading Basalts from the Nanyue Seamount: Implications for the Involvement of Crustal- and Plume-Type Components in the Genesis of the South China Sea Mantle"

_minerals, doi:10.3390/min9060378_

Round 1

Reviewer 1 Report

Review of Minerals manuscript 326075 v.1, “Post-spreading basalts from the Nanyue seamount: implications for the involvement of crustal- and plume-type components in the genesis of the South China Sea mantle” written by Zheng et al.

Overall Comments and Recommendation

This is a useful contribution to our understanding of the origin of basalts in the ocean basin. Although the paper does not make the point, seamounts do not receive the attention they deserve; their composition (such as in this paper) suggests that enriched mantle sources are more pervasive than just what we see at Oceanic Islands; so the paper is important from this perspective. The research is based on a small number of samples but they encompass the seamounts stratigraphy, it is unlikely that more samples would yield improved insights, and this seamount is likely to give a useful perspective on others from this tectonically-interesting area. The authors provide a whole-rock data set that is as comprehensive as you will find for individual samples in the literature and it includes major elements, extensive trace elements and Sr, Nd, Pb, Hf and Os isotopic ratios. This comprehensive amalgamation of data for individual samples facilitates testing of petrogenetic hypotheses, and represents an important resource for people who do “data mining”. The vast majority of basalt analyses in the literature never have as comprehensive a data set, yet each element and isotope provides information, and unthought-of combinations of data can be essential for testing some present or future hypotheses. Although I have a few gripes about data discussion, I think the paper provides useful insights into the origin of chemical variation in the mantle. Further the manuscript is well-organized and clearly written. That said, it is not succinct, is somewhat “wordy” and provides unnecessary details particularly in the Geological Background and Sampling and Petrography sections. These sections encompass 4 single-spaced pages of the manuscript to describe where 4 basalts came from and the petrography of most basalts, including these, is extremely simple. A final point is that the value of the fabulous data set has the potential to far outlive the data interpretations and I think the data deserve to be in the paper (not in an appendix).

I recommend publication pending minor to modest revisions.

The detailed comments below point out various minor wording issues, give examples of how the text can be shortened to improve impact and readability, and identify a few places where “interpretation” needs clarification. These detailed comments do not significantly address the “wordy” nature of the paper and in particular the Geological Background and Sampling and Petrography sections.

Detailed Comments

               Abstract

Line 15                 …enriched light rare…

L 17                       …shows they formed …

L18                        …and thus are products,,,

L26                        ...of the Hainan plume.

               Introduction

L44                        …domains with a Dupal-like…

               Geological Background

L81                        …plates, the SCS is regarded…

L93                        Continental rifting followed…

L94                        …era [e.g., 16, 18]. These…

               Sampling and Petrography

L171                     …collected at a shallow depth…

L175                     …with a yellow, unidentified secondary mineral forming amygdules …

L186                     Comment: the mesostasis is described as “amorphous”. Doe this mean it is vitreous,  devitrified glass, altered glass or so fine-grained it approaches isotropic but is not completely isotropic, or does amorphous mean something else. In other words can the authors provide a word or two about what the amorphous material is?

L187                     …locally containing …

L191                     Comment: the single twins may be Carlsbad twins but I suspect this is unnecessary detail as is the width of the twins.

Methods and Results

L215-L221           It is appropriate to give details of the analytical methods in the supplementary materials but I always feel that some perspectives on methods should be given in the text. Just a statement of the overall analytical methods is helpful. For example major elements by Atomic Absorption versus XRF on fused glass pellets with trace elements by emission spectroscopy versus ICP MS tells many readers a great deal. Most labs have a journal paper that describes their methods in detail. The references can be provided. Finally, there is an expectation that major and trace element analyses were run with replicates of a reference standard that allow precision and accuracy to be estimated, and run with a blank that provides detection limits. There are only 4 samples. The precision, accuracy and detection limits can easily be provided in 3 columns in the data table that contains the 4 sample analyses. As noted in the Overall Comments, I think that the data need to be provided in the main part of the paper.

L227-228             They have high sodium and elevated Na2O/K2O ratios…

L229                     …the compositions plot in the…

L232                     …values (1.51-1.82 wt%) consistent with…alteration-related minerals. (deleted the end of the sentence.

L238-239             …plotted on primitive mantle (PM)-normalized diagrams the basalts exhibit…

L247                     …basalts from near-by Hainan… You could also use …from the adjacent Hainan…

L260, L264, L270, L310, L347 and other places?   The geochemical data table that was put in the Appendix is referred to numerous times in the text. The table deserves to be in the text and this is especially so because there are only 4 samples.

L267-269             It is stated that the “composition of the Nanyue basats is unlikely to be explained by subduction of ancient pelagic sediment”. There are several problems here: 1) Pelagic sediment is not plotted on Figure 5 diagrams and it is not “obvious” to me how information on the diagram negates there being ancient pelagic sediment in the mantle source. This needs some explaining. 2) This sentence is interpretation and it (and an expansion of the sentence) belong in the Discussion. 3) Various explanations have been provided for EM signatures in OIB basalts and one is that they reflect ancient subducted oceanic sediment. Later in the paper it is argued that these rocks have an EM component in their source. If so, the origin of that EM signature may be subducted sediment. There are numerous, simple ways to deal with all of this but I leave it to the authors to sort it out.

L293                     …with a Dupal anomaly… This problem occurs other places. Search for Dupal.

L315                     …have 187Re/188Os ratios… These ratios are not shown in any figures or tables in the paper. If presented somewhere the reader should be referred to where they occur.

Discussion

L283                     …with a Dupal…

L391-L392           Comment: Maybe the authors are right and their basalts cannot represent SCLM because they have Os that is too radiogenic to have come from the SCLM and maybe the Os value they refer to for SCLM is representative. However, a problem we are all faced with is that the Subcontinental Lithospheric Mantle (SLM) is a major geochemical reservoir but 1) it is not clear that xenoliths provide representative samples of SLM because the mechanical properties of some lithologies may result in over-representation or under-representation in xenolith populations, 2) many studies based on xenoliths indicate that SLM is chemically stratified and stratification is different from one craton/locality to another, 3) the lower-most, hottest and least-brittle portions of SLM may not be sampled by xenoliths, but are the most-likely to be melted by asthenospheric magmas impinging on SLM and 4) based on magmas that apparently were affected by SLM, those that interacted with Archean SLM are very different from those that interacted with Proterozoic SLM (Greenough and McDivitt, 2018; reference below). In summary, the average composition and compositional variability of SLM is poorly constrained and the same goes for the Os isotopic composition of SLM. For example compare the Os isotopic composition of ancient SLM xenoliths with the composition of the host magma that melted Archean SLM in Carlson and Irving (1994). The host magma, which with other Wyoming magmas yields an Archean Pb-Pb model age (Greenough and Kyser 2003) has a completely different Os isotopic composition than the xenoliths; the host magma isotopic composition is hidden away as a footnote in one of the Carlson and Irving (1994) tables. There may be another problem with Os isotopes that has not received the attention it deserves. Cursory observations over the years are that Os isotopic ratios are sensitive to the percentage of melting due to disequilibrium melting. Re resides in Os-poor silicate phases and Os resides in Re-poor alloy phases that do not melt until high temperatures are reached (i.e. high percentages of melting). Thus low percentage melts tend to have radiogenic Os and the highest percentage melts end up with the highest Os concentrations but low, unradiogenic Os ratios. The overall point is that the Os isotopic composition of SLM is not well established and Os isotopes do not necessarily distinguish/identify SCLM. Well that is my opinion. Do with the above comments what you want and perhaps you want to stick with your argument on lines 391-392.

Greenough, J.D. and McDivitt, J.A. 2018. Earth’s evolving subcontinental lithospheric mantle: inferences from LIP continental flood basalt geochemistry. International Journal of Earth Sciences (Geol Rundsch) 107, pp. 787-810. DOI 10.1007/s00531-017-1493-6.

Carlson RW, Irving AJ (1994) Depletion and enrichment history of subcontinental lithospheric mantle: an Os, Sr, Nd, and Pb isotopic study of ultramafic xenoliths from the northwestern Wyoming craton. Earth Planet Sci Lett 126:457–472.

Greenough, J.D. and Kyser, T.K. 2003. Contrasting Archean and Proterozoic lithospheric mantle: Isotopic evidence from the Shonkin Sag sill (Montana). Contributions to Mineralogy and Petrology, v. 145, pp. 169-181.

L399                     …the near-by Hainan… or use adjacent.

L418                     The basalts studied…

L420                     …with those of [5], the…

L430                     …from a non-indigenous mantle… …case, the source could…

L438                     …the southeastern China…

L441                     …underlying the SCS.

L443                     …the Dupal isotopic anomaly is in the SCS… Check that this wording is appropriate.

L451                     …mantle with the Dupal…

L451-452             …beneath the Southwest sub-basin of the SCS concerns …time the original DM…

L464                     …from the neary-by and…

L490                     …the adjacent Hainan diaper.

L492                     …isotopic anomalies under some…

Author Response

To the

Editor-in-Chief

of Minerals (MDPI)

Dear Editor,

       As attached file to the present, you will find the revised version of the manuscript “Post-spreading basalts from the Nanyue seamount: implications for the involvement of crustal- and plume-type components in the genesis of the South China Sea mantle”, by Zheng et al. We are grateful to both reviewers for their constructive criticism and insightful comments. We reviewed our paper taking into account all their suggestions. As you will find out we made significant changes (marked with red color) to the following sections: Abstract, Introduction, Geological background, Sampling strategy, Petrography, Analytical methods and Discussion.

       A list with our reply to the reviewers’ queries, along with all the corrections we made, is given below.

Response to Reviewer 1 Comments

We did all the linguistic corrections the first reviewer asked us to do. So, below you will find our reply to the major comments made by the first reviewer.

Comment 1: the mesostasis is described as “amorphous”. Does this mean it is vitreous, devitrified glass, altered glass or so fine-grained it approaches isotropic but is not completely isotropic, or does amorphous mean something else. In other words can the authors provide a word or two about what the amorphous material is?

Response 1: We added the following information in the revised version of the text: the mesostasis is mainly composed of volcanic glass containing microlites of plagioclase, clinopyroxene and metallic minerals.

Comment 2: the single twins may be Carlsbad twins but I suspect this is unnecessary detail as is the width of the twins.

Response 2:We will agree with the reviewer. We removed the sentence about twinning in plagioclase from our text.

Comment 3: It is appropriate to give details of the analytical methods in the supplementary materials but I always feel that some perspectives on methods should be given in the text. Just a statement of the overall analytical methods is helpful. For example major elements by Atomic Absorption versus XRF on fused glass pellets with trace elements by emission spectroscopy versus ICP MS tells many readers a great deal. Most labs have a journal paper that describes their methods in detail. The references can be provided. Finally, there is an expectation that major and trace element analyses were run with replicates of a reference standard that allow precision and accuracy to be estimated, and run with a blank that provides detection limits. There are only 4 samples. The precision, accuracy and detection limits can easily be provided in 3 columns in the data table that contains the 4 sample analyses. As noted in the Overall Comments, I think that the data need to be provided in the main part of the paper.

Response 3: We added details about the analytical methods we used in the main text. A detailed description of all the analytical procedures we applied is given in the supplementary materials file. In the revised version of the supplementary materials file we cite at least one representative paper for the description of each analytical technique we used. Finally, we added analyses of reference materials and the blank in Table S1 in the supplementary materials file.

For the reason that we present a full range of analyzed elements (major, trace, REE, PGE) we think that we should better leave our tables in the supplementary materials file.

Comment 4: L260, L264, L270, L310, L347 and other places? The geochemical data table that was put in the Appendix is referred to numerous times in the text. The table deserves to be in the text and this is especially so because there are only 4 samples.

Response 4: In the revised version of the paper and especially in the section “Analytical methods” we note for one and only time that our compositional, isotopic and geochronological data are available in tables presented in the supplementary materials file.

Comment 5: It is stated that the “composition of the Nanyue basalts is unlikely to be explained by subduction of ancient pelagic sediment”. There are several problems here: 1) Pelagic sediment is not plotted on Figure 5 diagrams and it is not “obvious” to me how information on the diagram negates there being ancient pelagic sediment in the mantle source. This needs some explaining. 2) This sentence is interpretation and it (and an expansion of the sentence) belong in the Discussion. 3) Various explanations have been provided for EM signatures in OIB basalts and one is that they reflect ancient subducted oceanic sediment. Later in the paper it is argued that these rocks have an EM component in their source. If so, the origin of that EM signature may be subducted sediment. There are numerous, simple ways to deal with all of this but I leave it to the authors to sort it out.

Response 5: We corrected the diagram illustrated in figure 5b. Now, the line representing the isotopic trend for magmas produced by melting of old sediments is presented. We followed the reviewer’s suggestion and we moved our interpretations to the discussion. In the second chapter of our discussion we explain in detail why we think that pelagic or terrigenous sediments cannot sufficiently explain the isotopic signatures of basalts from the Nanyue seamount.

Comment 6: Maybe the authors are right and their basalts cannot represent SCLM because they have Os that is too radiogenic to have come from the SCLM and maybe the Os value they refer to for SCLM is representative. However, a problem we are all faced with is that the Subcontinental Lithospheric Mantle (SLM) is a major geochemical reservoir but 1) it is not clear that xenoliths provide representative samples of SLM because the mechanical properties of some lithologies may result in over-representation or under-representation in xenolith populations, 2) many studies based on xenoliths indicate that SLM is chemically stratified and stratification is different from one craton/locality to another, 3) the lower-most, hottest and least-brittle portions of SLM may not be sampled by xenoliths, but are the most-likely to be melted by asthenospheric magmas impinging on SLM and 4) based on magmas that apparently were affected by SLM, those that interacted with Archean SLM are very different from those that interacted with Proterozoic SLM (Greenough and McDivitt, 2018; reference below). In summary, the average composition and compositional variability of SLM is poorly constrained and the same goes for the Os isotopic composition of SLM. For example compare the Os isotopic composition of ancient SLM xenoliths with the composition of the host magma that melted Archean SLM in Carlson and Irving (1994). The host magma, which with other Wyoming magmas yields an Archean Pb-Pb model age (Greenough and Kyser 2003) has a completely different Os isotopic composition than the xenoliths; the host magma isotopic composition is hidden away as a footnote in one of the Carlson and Irving (1994) tables. There may be another problem with Os isotopes that has not received the attention it deserves. Cursory observations over the years are that Os isotopic ratios are sensitive to the percentage of melting due to disequilibrium melting. Re resides in Os-poor silicate phases and Os resides in Re-poor alloy phases that do not melt until high temperatures are reached (i.e. high percentages of melting). Thus low percentage melts tend to have radiogenic Os and the highest percentage melts end up with the highest Os concentrations but low, unradiogenic Os ratios. The overall point is that the Os isotopic composition of SLM is not well established and Os isotopes do not necessarily distinguish/identify SCLM. Well that is my opinion. Do with the above comments what you want and perhaps you want to stick with your argument on lines 391-392.

Response 6: We thank the first reviewer for his providing us with his detailed view on the Os-isotopic composition of the SCLM. The fact that we do not have many Os-isotopic data from the investigated basalts does not allow us to reach a “safer” petrogenetic conclusion regarding the role of the subcontinental lithospheric mantle in controlling the Os-isotopic composition of post-spreading basalts in the broader South China Sea region.

We really appreciate the fact that the reviewer shared with us his ideas about the average compositional variability of SLM. The truth is that our paper represents part of a much bigger and ambitious project regarding the genesis and petrologic evolution of the mantle below the South China Sea. So, we would like to share with the reviewer the information that our preliminary Os-isotopic data of basalts from other seamounts of the South China Sea collectively show that the low concentrations of Os and the radiogenic Os-isotopic ratios could be interpreted as a result of fractionation of Os-bearing minerals at an early stage in the evolution of the parental melts of basalts. Finally, we have added the useful references the reviewer suggested.

Thank you very much for your time and consideration,

On behalf of all authors Dr. Li-Feng Zhong

Guangzhou, June 2018

Reviewer 2 Report

This manuscript documents petrographic and geochemical characteristics of basalts from the Nanyue seamount, South China Sea. The main conclusion presented in the manuscript is that upper mantle source during post-spreading magmatism at South China Sea has Dupal-like signature, which is produced by mixing of Pacific-type MORB mantle, LCC and the Hainan OIB components. This conclusion is an important contribution to understanding post-spreading magmatism at and source mantle characteristics beneath South China Sea.

 But, I'd like the authors to clarify the followings.

(1) Identification of samples, for example, the depth of sample recovered, macroscopic description of “each” sample (because all samples were recovered from different sites, it is plausible that they originated from different lava).

(2) There are some inappropriate interpretations and discussions.

 I believe that moderate revision can bring this manuscript to the level appropriate for journal. I list my comments that I think are helpful for the correction. They are mostly organization and presentation issues, and I find no serious flaw in logic or methods. However, I stress that serious revision is still needed. As such, I do not exclude revisions not listed in my comments. I am not a native English speaker, but the language itself seems overall acceptable, except a few obvious typos.

1. Line 94: Insert a period after “[e.g., 16, 18]”

2. Line 101: In Figure 1b, should text “Northwest sub-basin” move to lower right? Because “Northwest sub-basin” is a part of the ocean basin (light-blue color in Figure 1b, I guess). Furthermore, it is easy for readers to understand altitude or water depth if you put a color scale as a legend. 

3. Line 128-132: In figure 1b, I can’t find the Spartly Islands and Changlong, Longnan, Longbei, and Zhongnan seamounts. Detailed bathymetric map of the Southwest sub-basin helps the readers to understand the location and relationship of seamounts and seamount chain. Is “Changlong" the name of a seamount, seamount chain, or both? Is "Spartly"  typo of Spratly?

4. Line 155-156: The authors mention that “A total of 8 rock specimens were taken from different sites of the flanks of the Nanyue seamount”. How many sites was dredging performed? Were all 8 rocks specimens taken from eight different sites? 

5. Line 179-: In the following section, the authors describe microscopic (petrographic) descriptions for six fresh volcanic rocks. Were described six rocks recovered from the same dredging site? If not, because there is a possibility that they represent different magmatic activities, the authors should describe for each sample. If the rocks were from the same dredging site, the authors must change sentence at line 155-156.

6. Line 188-189: Are these samples free from olivine phenocryst? Further, how many samples did the authors examine modal analyses?  

7. Line 216: Are there any reason why four samples were chosen? The authors mention "2 volcanic rock samples … were excluded from further examination" at Line 157. Because the total number of sample is 8, the author examined 6 samples. The authors should mention a reason or criterion to choose four samples from 6 examined samples. Were the four samples taken from different dredging sites or the same site?

8. Line 246-247: Because trace element profiles of Hainan basalt show wide distributions, it is difficult to mention "resembles" simply. 

9. Line 262-263: The authors describe that "the isotopic compositions of the investigated basalts straddle the fields of the Indian Ocean MORB and the Hainan Island OIB," but "straddle" is not suitable because in Figure 5 the field of the Hainan Island OIB overlaps with the Indian Ocean MORB.

10. Line 266-267: At least in Figure 5b, the compositions of the investigating basalts are the same (or similar) compositions to the Hainan OIB.

11. Line 273-274: I guess that "they" in line 273 indicates basalts from the East and Northwest sub-basins. If so, who describes that "they are analogous to those of MORB from the central Southwest Indian ridge?" Janney et al. (2005), which the authors referred did not mention the similarity between SCS basalts and SWIR MORB. 

12. Line 278: Is “In the 207Pb/204Pb versus 87Sr/86Sr” typo of “206Pb/204Pb versus 87Sr/86Sr”?

13. Line 327: Please add "(AP)" after "Data for abyssal peridotites." 

14. Line 329-332: Depth (or location) where samples were recovered should be described in section 3.1. Although the authors describe that “HD66-1 is from upper section and HD66-2 is from lower section”, which part did other samples come? 

15. Line 351-355: The authors discussed the possibility of crustal contamination based on Nb/Ta. But the following line 356-362, they also describe that Os-isotope data are not in favor of contamination in spite of Os-isotope is a typical "hallmark." Because the authors wrote "affected by crustal contamination to some extent," it is certain that the authors believe the possibility of crustal contamination. If so, the author should explain the reason why Os-isotope data don’t indicate the contamination, but Nb/Ta does.

16. Line 409-411: Can mixing of (most likely less than 50%) EM-type components from the Hainan plume with the “DM end-member” explain the Hf-Nd isotope systematics (Figure 5b)? It looks like that “DM end-member” does not have any contributions for Hf isotope.

17. Line 435: Add "Sea" as "Indo-Australia and Philippine Sea plates."

18. Line 470-471: Because 5.8 Ma seamount exists in the Northwest sub-basin (Figure 1b), the tendency of age which become younger from NW to SW sub-basin previously mentioned by ref. 34 “is not consistent” with this paper’s data.

19. Line 487-490: The authors concluded in conclusion 3 that Nanyue basalts were from an Indian-type upper mantle source. But as they discussed in section 5.2 and concluded in conclusion 4, the basalts were from the mixing Pacific-type MORB mantle+LCC and the Hainan component. This mixed component is similar to “Indian-type source” but is not Indian-type source itself based on the authors' discussions. If so, conclusion 3 can be misleading.

Author Response

To the

Editor-in-Chief

of Minerals (MDPI)

Dear Editor,

       As attached file to the present, you will find the revised version of the manuscript “Post-spreading basalts from the Nanyue seamount: implications for the involvement of crustal- and plume-type components in the genesis of the South China Sea mantle”, by Zheng et al. We are grateful to both reviewers for their constructive criticism and insightful comments. We reviewed our paper taking into account all their suggestions. As you will find out we made significant changes (marked with red color) to the following sections: Abstract, Introduction, Geological background, Sampling strategy, Petrography, Analytical methods and Discussion.

       A list with our reply to the reviewers’ queries, along with all the corrections we made, is given below.

 Response to Reviewer 2 Comments

       Comment 1: Line 94: Insert a period after “[e.g., 16, 18]”.

       Response 1: We inserted the period.

Comment 2: In Figure 1b, should text “Northwest sub-basin” move to lower right? Because “Northwest sub-basin” is a part of the ocean basin (light-blue color in Figure 1b, I guess). Furthermore, it is easy for readers to understand altitude or water depth if you put a color scale as a legend. 

Response 2: We moved the label “Northwest sub-basin” to the right to fall on the oceanic part of the basin. We also added a vertical scale to show the altitude in the continental regions around the SCs and the depth of water within the SCS.

Comment 3: In figure 1b, I can’t find the Spartly Islands and Changlong, Longnan, Longbei, and Zhongnan seamounts. Detailed bathymetric map of the Southwest sub-basin helps the readers to understand the location and relationship of seamounts and seamount chain. Is “Changlong" the name of a seamount, seamount chain, or both? Is "Spartly" typo of Spratly?

Response 3: We added the names of the Changlong, Longnan, Longbei, and Zhongnan seamounts on the map illustrated on figure 1b. We made the necessary correction to the name Spratly Islands. The name Changlong corresponds to a volcanic seamount and a seamount chain at the same time. It is one of the biggest seamounts in the southwest sub-basin of the SCS and the seamount chain that crosses the investigated sub-basin was named after the homonymous intraplate volcano. We do not think that adding an extra map would benefit the paper. The addition of an altimetry-bathymetry scale and the names of the volcanic seamounts that were missing to the map illustrated in figure 1b allow the reader to have access to the information he/she needs to understand the physiographic and geological characteristics of the study area.     

Comment 4: The authors mention that “A total of 8 rock specimens were taken from different sites of the flanks of the Nanyue seamount”. How many sites was dredging performed? Were all 8 rocks specimens taken from eight different sites?

Response 4: Our samples were taken from two sites. A relatively shallow site located at a depth of ~1020 m below the surface of water and another one located at a depth of ~1270 m below the surface of the sea. Both sites represent lava breakout sites and are located on the flanks of the investigated seamount. We took 4 basalt samples from the shallowest site and 4 basalt samples from the deepest site. We note that each site occupies an area of a few m2.  

Comment 5: In the following section, the authors describe microscopic (petrographic) descriptions for six fresh volcanic rocks. Were described six rocks recovered from the same dredging site? If not, because there is a possibility that they represent different magmatic activities, the authors should describe for each sample. If the rocks were from the same dredging site, the authors must change sentence at line 155-156.

Response 5: We describe in detail the macroscopic and microscopic features of the basalt specimens we collected from the two sampling sites on the flanks of the Nanyue seamount. We describe them as they form a single group of rocks. However, we note any significant structural, mineralogical and or microtextural differences among the studied samples whenever they occur. These differences give us a hint about the origin of the investigated basalts. Though the investigated basalts do not show any remarkable compositional or isotopic differences they have a different age. So, we will agree with the reviewer that basalts from each site might represent different lava flows.

Comment 6: Are these samples free from olivine phenocryst? Further, how many samples did the authors examine modal analyses?  

       Response 6: Our detailed petrographic examination showed that the investigated basalts are deprived of olivine crystals. The modal amounts of every mineral constituent in the Nanyue basalts were calculated using visual estimation methods.

Comment 7: Are there any reason why four samples were chosen? The authors mention "2 volcanic rock samples … were excluded from further examination" at Line 157. Because the total number of sample is 8, the author examined 6 samples. The authors should mention a reason or criterion to choose four samples from 6 examined samples. Were the four samples taken from different dredging sites or the same site?

       Response 7: We chose to analyze two representative samples from the deepest site. We noticed that one sample presented a different microstructure. So, we chose to analyze two samples with different structures from the deepest site. We had only two unaltered basalt samples from the shallowest site. The truth is that the analyzed samples from the shallowest site did not show any noteworthy petrographic differences.

Comment 8: Because trace element profiles of Hainan basalt show wide distributions, it is difficult to mention "resembles" simply. 

Response 8: We agree with the reviewer. Therefore, we replaced the term “resembles” with the term “follows”.

Comment 9: The authors describe that "the isotopic compositions of the investigated basalts straddle the fields of the Indian Ocean MORB and the Hainan Island OIB," but "straddle" is not suitable because in Figure 5 the field of the Hainan Island OIB overlaps with the Indian Ocean MORB.

Response 9: We will agree with the reviewer again. In the revised version of the paper we note that the isotopic compositions of the investigated rocks plot in the field of Indian MORB and in the overlapping area between the Indian MORB and the Hainan OIB.

Comment 10: At least in Figure 5b, the compositions of the investigating basalts are the same (or similar) compositions to the Hainan OIB.

       Response 10: Yes, this is true. A detailed explanation for the absence of isotopic decoupling between Nd and Hf is provided in the discussion. We believe that the effect of the “DM end member” on the Hf-isotopic signature of the studied rocks was soon erased by magma mixing processes.    

Comment 11: I guess that "they" in line 273 indicates basalts from the East and Northwest sub-basins. If so, who describes that "they are analogous to those of MORB from the central Southwest Indian ridge?" Janney et al. (2005), which the authors referred, did not mention the similarity between SCS basalts and SWIR MORB. 

Response 11: We made the necessary linguistic corrections. We also replaced the reference Janney et al. with Meyzen et al. 2005. 

Comment 12: Is “In the 207Pb/204Pb versus 87Sr/86Sr” typo of “206Pb/204Pb versus 87Sr/86Sr”?

Response 12: We could not find the mistake the reviewer is talking about.

Comment 13: Please add "(AP)" after "Data for abyssal peridotites." 

Response 13: We did it.

Comment 14: Depth (or location) where samples were recovered should be described in section 3.1. Although the authors describe that “HD66-1 is from upper section and HD66-2 is from lower section”, which part did other samples come? 

       Response 14: A detailed description of the depth and the sites our samples were collected from is provided in section 3.1 in the revised version of the manuscript.

Comment 15: The authors discussed the possibility of crustal contamination based on Nb/Ta. But the following line 356-362, they also describe that Os-isotope data are not in favor of contamination in spite of Os-isotope is a typical "hallmark." Because the authors wrote "affected by crustal contamination to some extent," it is certain that the authors believe the possibility of crustal contamination. If so, the author should explain the reason why Os-isotope data don’t indicate the contamination, but Nb/Ta does.

Response 15: We think that the investigated basalts do not show any evidence of affection of their Os-isotopic compositions by crustal contamination. In fact, the basalt sample (HD66-2) with the lowest Os concentration (60.37 ppt) and the highest 187Os/188Os (0.1856), has the lowest 87Sr/86Sr (0.704222) and εNd (4.0) of all the investigated samples, which would not be the case if crustal contamination was responsible for its Os-isotopic composition. In addition, the lack of a co-variation between the Os content and other trace element contents or radiogenic isotopes (that can be affected by crustal contamination) implies that the Os-isotopic composition of the Nanyue basalts is most likely the original one.

Comment 16: Can mixing of (most likely less than 50%) EM-type components from the Hainan plume with the “DM end-member” explain the Hf-Nd isotope systematics (Figure 5b)? It looks like that “DM end-member” does not have any contributions for Hf isotope.

Response 16: Decoupling of Hf from Nd is a known process during mantle melting. However, the Nanyue basalts originate from melts that were not characterized by this kind of geochemical decoupling. As we explained in one of the previous comments (10) we believe that the impact of the “DM end-member” on the Hf-isotopic signature of the Nanyue basalts was soon erased by mixing of the “DM end-member” with OIB-type melts. This is evidenced by: i) the fact that the upper mantle below the SCS has been influenced by plume-type melts, ii) the zoning texture preserved in plagioclase, and iii) the existence of clinopyroxenes with partially resorbed boundaries.

Comment 17: Add "Sea" as "Indo-Australia and Philippine Sea plates."

Response 17: We did it.

Comment 18: Because 5.8 Ma seamount exists in the Northwest sub-basin (Figure 1b), the tendency of age which become younger from NW to SW sub-basin previously mentioned by ref. 34 “is not consistent” with this paper’s data.

Response 18: We agree with the reviewer. In the revised version of the paper we emphasize that our data are not consistent with the argument of the ageing of the seamounts from the south to the north. We also note that there is no direct relationship between the geochemistry and age of post-spreading volcanism in the Southwest sub-basin of the SCS.   

Comment 19: The authors concluded in conclusion 3 that Nanyue basalts were from an Indian-type upper mantle source. But as they discussed in section 5.2 and concluded in conclusion 4, the basalts were from the mixing Pacific-type MORB mantle+LCC and the Hainan component. This mixed component is similar to “Indian-type source” but is not Indian-type source itself based on the authors' discussions. If so, conclusion 3 can be misleading.

       Response 19: The term Indian-type upper mantle source in the third conclusion was replaced by the term upper mantle source with a Dupal-like isotopic anomaly.

Thank you very much for your time and consideration,

On behalf of all authors Dr. Li-Feng Zhong

Guangzhou, June 2018
